



An updated synthesis of ocean total alkalinity and dissolved inorganic carbon measurements
from 1993 to 2023: the SNAPO-CO2-v2 dataset
Nicolas Metzl[1], Jonathan Fin[1,2,3], Claire Lo Monaco[1], Claude Mignon[1], Samir Alliouane[4], Bruno Bombled[5],
Jacqueline Boutin[1], Yann Bozec[6], Steeve Comeau[4], Pascal Conan[7,8], Laurent Coppola[4,8], Pascale Cuet[9], Eva
Ferreira[5], Jean-Pierre Gattuso[4,10], Frédéric Gazeau[4], Catherine Goyet[11], Emilie Grossteffan[12], Bruno Lansard[5],
Dominique Lefèvre[13], Nathalie Lefèvre[1], Coraline Leseurre[14], Sébastien Petton[15], Mireille Pujo-Pay[8], Christophe
Rabouille[5], Gilles Reverdin[1], Céline Ridame[1], Peggy Rimmelin-Maury[12], Jean-François Ternon[16], Franck
Touratier[11], Aline Tribollet[1], Thibaut Wagener[13], Cathy Wimart-Rousseau[17].
[1] Laboratoire LOCEAN/IPSL, Sorbonne Université-CNRS-IRD-MNHN, Paris, 75005, France
[2] OSU Ecce Terra, Sorbonne Université-CNRS, Paris, 75005, France
[3] Now at Institut des Sciences de la Terre, Grenoble, 38058, France
[4] Sorbonne Université, CNRS, Laboratoire d'Océanographie de Villefranche, LOV, F-06230 Villefranche-sur-
Mer, France
[5] Laboratoire des Sciences du Climat et de l'Environnement, LSCE/IPSL, UMR 8212 CEA- CNRS-UVSQ,
Université Paris-Saclay, 91191 Gif-sur-Yvette, France
[6] Station Biologique de Roscoff, UMR 7144 – EDYCO-CHIMAR, Roscoff, France
[7] Sorbonne Université, CNRS, Laboratoire d'Océanographie Microbienne, LOMIC, F-66650 Banyuls-sur-Mer,
France
[8] Sorbonne Université, CNRS OSU STAMAR - UAR2017, 4 Place Jussieu, 75252, Paris cedex 05, France
[9] Laboratoire ENTROPIE and Laboratoire d'Excellence CORAIL, Université de La Réunion-IRD- CNRS-
IFREMER-Université de la Nouvelle-Calédonie, 97744, Saint‑Denis, La Réunion, France
[10] Institute for Sustainable Development and International Relations, Sciences Po, 27 rue Saint Guillaume, F-
75007 Paris, France
[11] Espace-Dev UMR 228 Université de Perpignan Via Domitia, IRD, UM, UA, UG, 66860, Perpignan, France
[12] Institut Universitaire Européen de la Mer (OSU-IUEM), Univ Brest, CNRS-UAR3113, 29280, Plouzané,
France
[13] Aix Marseille Univ, Université de Toulon, CNRS, IRD, MIO, Marseille, France
[14] Flanders Marine Institute (VLIZ), 8400 Ostend, Belgium
[15] Ifremer, Univ Brest, CNRS, IRD, LEMAR, F-29840 Argenton, France
[16] MARBEC, Univ Montpellier, CNRS, Ifremer, IRD, Sète, France
[17] National Oceanography Centre Southampton, European Way, Southampton, SO14 3ZH, UK
*Correspondence to*: Nicolas Metzl (nicolas.metzl@locean.ipsl.fr)
**Abstract**. Total alkalinity ($A_T$) and dissolved inorganic carbon ($C_T$) in the oceans are important properties to
understand the ocean carbon cycle and its link with global change (ocean carbon sinks and sources, ocean
acidification) and ultimately find carbon based solutions or mitigation procedures (marine carbon removal). We
present an extended database (SNAPO-CO2, Metzl et al, 2024d) with 24700 new additional data for the period
2002 to 2023. The full database now includes more than 67000 $A_T$ and $C_T$ observations along with basic
ancillary data (time and space location, depth, temperature and salinity) in various oceanic regions obtained since
1993 mainly in the frame of French research projects. This includes both surface and water columns data
acquired in open oceans, coastal zones, rivers and in the Mediterranean Sea and either from time-series or
punctual cruises. Most $A_T$ and $C_T$ data in this synthesis were measured from discrete samples using the same
closed-cell potentiometric titration calibrated with Certified Reference Material, with an overall accuracy of ± 4
µmol kg$^{-1}$ for both $A_T$ and $C_T$. The same technique was used onboard for underway measurements during cruises





conducted in the Southern Indian and Southern Oceans. The $A_T$ and $C_T$ data from these cruises are also added in
this synthesis. The data are provided in one dataset for the global ocean (https://doi.org/10.17882/102337) that
offers a direct use for regional or global purposes, e.g. $A_T$/Salinity relationships, long-term $C_T$ estimates,
constraint and validation of diagnostics $C_T$ and $A_T$ reconstructed fields or ocean carbon and coupled
climate/carbon models simulations, as well as data derived from Biogeochemical-Argo (BGC-Argo) floats.
These data can also be used to calculate pH, fugacity of $CO_2$ ($fCO_2$) and other carbon system properties to derive
ocean acidification rates or air-sea $CO_2$ fluxes.

**1 Introduction**

59        The ocean plays a major role in reducing the impact of climate change by absorbing more than 90% of

the excess heat in the climate system (Cheng et al., 2020, 2024; von Schuckmann et al, 2023; IPCC, 2022) and
about 25% of human released $CO_2$ (Friedlingstein et al., 2022, 2023). In the last decade, the oceans experienced
a rapid warming, the year 2023 being the hottest since 1955 (Cheng et al, 2024). In the atmosphere the $CO_2$
concentration continues its terrific progressive rising, reaching 419.3 ppm in 2023 (a rate of +2.83 ppm yr$^{-1}$, Lan
et al 2024). In August 2024, the global atmospheric $CO_2$ concentration was already above 420 ppm. In the next
decade the oceans will continue to capture heat and $CO_2$, somehow limiting the climate change, but this oceanic
$CO_2$ uptake changes the chemistry of seawater reducing its buffering capacity (Revelle and Suess, 1957; Jiang et
al, 2023). This process known as ocean acidification has potential impacts on marine organisms (Fabry et al.,
2008; Doney et al., 2009, 2020; Gattuso et al., 2015). With atmospheric $CO_2$ concentrations, surface ocean
temperature and ocean heat content, sea-level, sea-ice and glaciers, the ocean acidification (decrease of pH) is
now recognized by the World Meteorological Organization as one of the 7 key properties for global climate
indicators (WMO, 2018). Ocean acidification is specifically referred in the SDG indicator 14.3.1 coordinated at
the Intergovernmental Oceanographic Commission (IOC) of UNESCO. Observing the carbonate system in the
open oceans, coastal zones and marginal seas and understanding how this system changes over time is thus
highly relevant not only to quantify the global ocean carbon budget, the anthropogenic $CO_2$ inventories or ocean
acidification rates, but also to understand and simulate the processes that govern the complex $CO_2$ cycle in the
ocean (e.g. Goyet et al, 2016, 2019) and to better predict the future evolution of climate and global changes
(Eyring et al., 2016; Kwiatkowski et al., 2020; Jiang et al., 2023). As the rate of change in ocean acidification
presents large temporal and regional variability, long-term observations are required. Weekly to monthly regular
resolution data are needed to better investigate the long-term change of the carbonate system in regions subject
to extreme events (e.g. tropical cyclones, marine heat or cold waves, rapid freshening, convection, dust events,
river discharges, etc....). In this context it is recommended to progress in data synthesis of the ocean carbon
observations that would offer new high quality products for the community (e.g. for GOA-ON, www.goa-on.org,
IOC/SDG 14.1.3, https://oa.iode.org/, Tilbrook et al., 2019).

84        In this work, following the first SNAPO-CO2 synthesis product (Metzl et al, 2024a), we present a new

synthesis of more than 67000 $A_T$ and $C_T$ data, measured either on shore or onboard Research Vessels obtained
over the 1993-2023 period during various cruises or at time-series stations mainly supported by French projects.
Hereafter this new dataset will be cited as SNAPO-CO2-v2. The methods, data assemblage and quality control





were presented in version V1. Here, we describe the new data added and discuss some potential uses of this
dataset.
**2 Data collections**
93       The time series projects and research cruises from which new data were collated are listed in Table 1
with information and references in the Supplementary file (Tables S1, S3 and S4). The sampling locations of
new data are displayed in Figure 1 (the location for all data presented in Figure S1). Sampling was performed
either from CTD-Rosette casts (Niskin bottles) or from the ship's seawater supply (intake at about 5m depth
depending on the ship and swell). Samples collected in 500 mL borosilicate glass bottles were poisoned with 100
to 300 μL of $HgCl_2$ depending on the cruises, closed with greased stoppers (Apiezon®) and held tight using
elastic band following the SOP protocol (DOE, 1994; Dickson et al., 2007). Some samples were also collected in
500 mL bottles closed with screw caps. After completion of each cruise, most of discrete samples were returned
back to the LOCEAN laboratory (Paris, France) and stored in a dark room at 4 °C before analysis generally
within 2-3 months after sampling (sometimes within a week). In this version we added data from samples that
were also returned to University of Perpignan or to University of La Réunion. In addition to discrete samples
analyzed for various projects conducted mainly in the North Atlantic, Tropical Atlantic, Mediterranean Sea and
coastal regions (Table 1), we complemented this second synthesis with $A_T$ and $C_T$ surface observations obtained
in the Indian and Southern oceans during the OISO cruises in 2019-2021 (Leseurre et al., 2022; Metzl et al,
2022; data also available at NCEI/OCADS: www.nodc.noaa.gov/ocads/oceans/VOS_Program/OISO.html) and
MINERVE cruises in 2002-2018 (Laika et al, 2009; Brandon et al, 2022). The $A_T$ and $C_T$ measurements from the
MINERVE cruises were performed either onboard R/V Astrolabe or back in the laboratories (at LOCEAN
laboratory and at University of Perpignan).

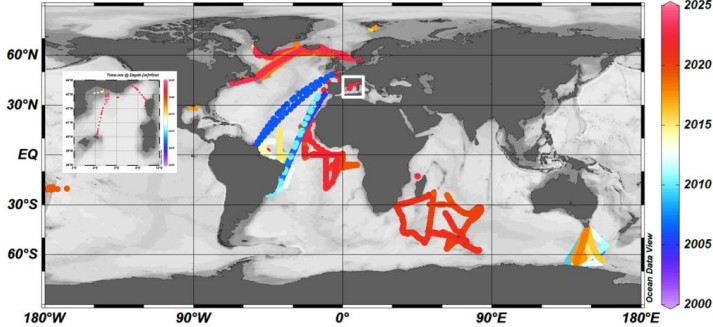

122  **Figure 1:** Locations of new $A_T$ and $C_T$ data (2005-2023) in the Global Ocean and the Western Mediterranean Sea
123  (white box, insert) in the SNAPO-CO2-v2 dataset. Color code is for Year. Figure produced with ODV (Schlitzer,
124  2018).
125

**Table 1:** List of cruises added in the SNAPO-CO2-v2 dataset. This is organized by region from North to South and the Mediterranean Sea. See Tables S1, S2, S3 and S4 in the Supplementary Material for a list of laboratories, of CRMs used, for DOI and for references of cruises. Nb = the number of data for each cruise or time-series. * indicates the measurements at sea (surface underway).

| Cruise/Project | Start | End | Region | Sampling | Nb |
|---|---|---|---|---|---|
| STEP | 2016 | 2017 | Arctic | Water Column | 33 |
| SURATLANT AX1 | 2017 | 2023 | North Atlantic | Surface | 255 |
| SURATLANT AX2 | 2018 | 2023 | North Atlantic | Surface | 224 |
| VOS | 2005 | 2010 | Atlantic | Surface | 192 |
| MISSRHODIA-1 | 2017 | 2017 | Gulf Mexico | Water Column | 8 |
| ACIDHYPO | 2022 | 2022 | Gulf Mexico | Water Column | 10 |
| CAMFIN-WATL | 2010 | 2015 | Trop Atlantic | Surface | 192 |
| PIRATA-BR | 2009 | 2015 | Trop Atlantic | Surface | 194 |
| BIOAMAZON | 2013 | 2014 | Trop Atlantic | Surface | 62 |
| AMAZOMIX | 2021 | 2021 | Trop Atlantic | Water Column | 180 |
| PIRATA-FR | 2019 | 2019 | Trop Atlantic | Surface | 93 |
| PIRATA-FR | 2020 | 2020 | Trop Atlantic | Surface, Water Column | 58 |
| PIRATA-FR | 2021 | 2021 | Trop Atlantic | Surface, Water Column | 79 |
| PIRATA-FR | 2022 | 2022 | Trop Atlantic | Surface, Water Column | 118 |
| CO2ARVOR | 2009 | 2010 | Atlantic, Coastal | Surface, Water Column | 621 |
| SOMLIT-Roscoff | 2020 | 2022 | Coastal North Atl | Surface and 60m | 207 |
| SOMLIT-Brest | 2020 | 2022 | Coastal North Atl | Surface | 251 |
| TONGA | 2019 | 2019 | Trop Pacific | Water Column | 226 |
| CARBODISS | 2018 | 2019 | Indian Mayotte | Surface | 85 |
| OISO * | 2019 | 2021 | South Indian | Surface | 5258 |
| MINERVE | 2004 | 2018 | Southern Ocean | Surface | 1077 |
| MINERVE * | 2002 | 2013 | Southern Ocean | Surface | 11258 |
| COCORICO2 | 2017 | 2022 | Coastal | Surface | 589 |
| SOMLIT-PointB | 2019 | 2023 | MedSea Coastal | Surface and 50m | 716 |
| SOLEMIO | 2018 | 2022 | MedSea Coastal | Water Column | 271 |
| ANTARES | 2017 | 2023 | MedSea | Water Column | 506 |
| MOLA | 2018 | 2023 | MedSea Coastal | Water Column | 193 |
| DYFAMED | 2018 | 2023 | MedSea | Water Column | 514 |
| MESURHO-BENT | 2010 | 2011 | MedSea Coastal | Surface and sub-surface | 25 |
| ACCESS-01 | 2012 | 2012 | MedSea Coastal | Water Column | 16 |
| CARBO-DELTA-2 | 2013 | 2013 | MedSea Coastal | Water Column | 14 |
| DICASE | 2014 | 2014 | MedSea Coastal | Water Column | 22 |
| MISSRHODIA-2 | 2018 | 2018 | MedSea Coastal | Surface and sub-surface | 13 |
| DELTARHONE1 | 2022 | 2022 | MedSea Coastal | Water Column | 9 |
| MOOSE-GE | 2021 | 2021 | MedSea | Water Column | 451 |
| MOOSE-GE | 2022 | 2022 | MedSea | Water Column | 447 |
| MOOSE-GE | 2023 | 2023 | MedSea | Water Column | 475 |

## 3 Method, accuracy, repeatability and quality control

### 3.1 Method and accuracy

Since 2003, the discrete samples returned back at SNAPO-CO2 Service facilities (LOCEAN, Paris), were analyzed simultaneously for $A_T$ and $C_T$ by potentiometric titration using a closed cell (Edmond, 1970; Goyet et al., 1991). The same technique was used at sea for surface water underway measurements during OISO and MINERVE cruises (indicated by * in Table 1). In the late 1980s the so-called "JGOFS-IOC Advisory Panel on Ocean CO2" recommended the need for standard analysis protocols and for developing Certified Reference Materials (CRMs) for inorganic carbon measurements (Poisson et al., 1990; UNESCO, 1990, 1991). The CRMs



were provided to international laboratories by Pr. A. Dickson (Scripps Institution of Oceanography, San Diego,
USA), starting in 1990 for $C_T$ and 1996 for $A_T$, respectively. These CRMs were thus always available to us and
used to calibrate the measurements (CRM Batch numbers used for each cruise are listed in the Supplementary
file, Table S2). The CRMs accuracy, as indicated in the certificate for each Batch, is around ±0.5 µmol kg$^{-1}$ for
both $A_T$ and $C_T$ (www.nodc.noaa.gov/ocads/oceans/Dickson_CRM/batches.html). The concentrations of CRMs
we used vary between 2193 and 2426 µmol kg$^{-1}$ for $A_T$ and between 1968 and 2115 µmol kg$^{-1}$ for $C_T$
corresponding to the range of concentrations observed in open ocean water. In the Mediterranean Sea the
concentrations are higher ($A_T$ > 2600 µmol kg$^{-1}$ and $C_T$ > 2300 µmol kg$^{-1}$) and in the coastal zones or near the
Amazon River plume the concentrations were often lower than the CRMs ($A_T$ < 1500 µmol kg$^{-1}$ and $C_T$ < 1000
µmol kg$^{-1}$). Results of analyses performed on 1242 CRM bottles (different Batches) in 2013-2024 are presented
in Figure 2. The standard-deviations (Std) of the differences of measurements were on average ±2.69 µmol kg$^{-1}$
for $A_T$ and ±2.88 µmol kg$^{-1}$ for $C_T$. For unknown reasons, the differences were occasionally up to 10-15 µmol kg$^{-1}$
(1.2% of the data, Figure S2). These few CRM measurements were discarded for the data processing. We did
not detect any specific signal for CRM analyses (e.g., larger uncertainty depending on the Batch number or
temporal drifts during analyses, Figure 2) but for some cruises the accuracy based on CRMs could be better than
3 µmol kg$^{-1}$ (e.g. < 3 µmol kg$^{-1}$ for AMAZOMIX cruise using 6 Batches #197 and for MOOSE-GE 2022 using
19 Batches #204, or < 1.5 µmol kg$^{-1}$ for SOMLIT-Point-B in 2022 using 6 Batches #204).

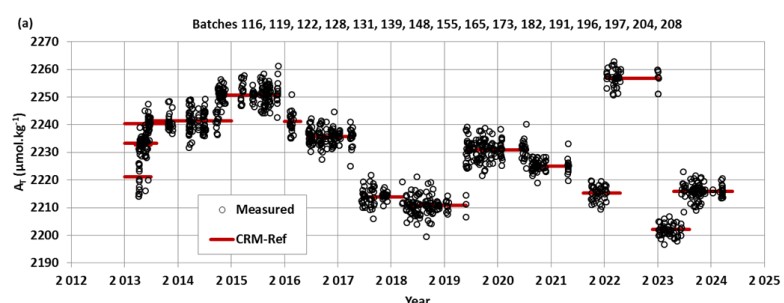

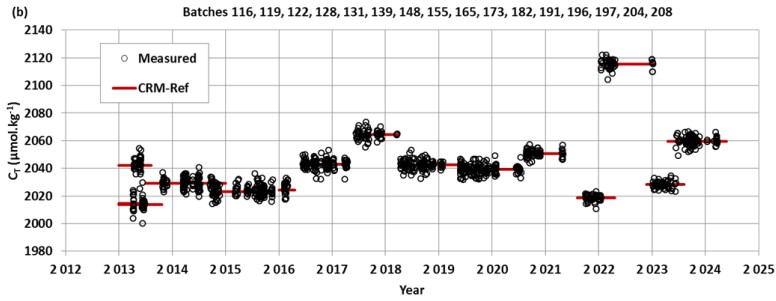

**Figure 2:** $A_T$ (a) and $C_T$ (b) analyses for different CRM Batches measured in 2013-2024. For these 1242 analyses the mean and standard-deviations of the differences with the CRM reference were -0.11 (± 2.69) µmol kg$^{-1}$ for $A_T$ and 0.01 (± 2.88) µmol kg$^{-1}$ for $C_T$.
**3.2 Repeatability**

For some projects, duplicates have been regularly sampled (SOMLIT-Point-B, SOMLIT-Brest) or replicate bottles sampled at selected depths at fixed stations during the cruises (e.g. STEP, CARBODISS). In the first synthesis of the SNAPO-CO2 dataset we showed the results from several time-series (SOMLIT-Point-B, SOMLIT-Brest and BOUSSOLE/DYFAMED). Here we present the results for the new data obtained at SOMLIT-Point-B in the coastal Mediterranean Sea and SOMLIT-Brest in the Bay of Brest (Figure 3). Results of $A_T$ and $C_T$ repeatability are synthetized in Table 2. For the OISO cruises conducted in 2019, 2020 and 2021 the repeatability was evaluated from duplicate analyses (within 20 minutes time) of continuous sea surface underway sampling at the same location (when the ship was stopped). Similarly to what was found for the CRM measurements (Figure S2), differences in duplicates are occasionally higher than 10-15 µmol kg$^{-1}$ (Figure 3) but most of the duplicates for all projects are within 0 to 3 µmol kg$^{-1}$. Compared to previous results (Kapsenberg et al. 2017; Metzl et al, 2024a), there are larger differences between duplicates at SOMLIT-B in 2019-2023 (up to 30 µmol kg$^{-1}$, Figure 3) leading to relatively large Std around 5 and 6 µmol kg$^{-1}$ for both $A_T$ and $C_T$ (Table 2). The same was observed for duplicates at SOMLIT-Brest (Table 2). We do have not yet a clear explanation for this large Std although larger variability was observed in recent years, and the measurements were performed later after the sampling (e.g. more than 6 months for some samples during and after the COVID period). We will see that given the temporal variability of the properties this does not lead to suspicious interpretation for the seasonality or the trend analyses of these time-series.

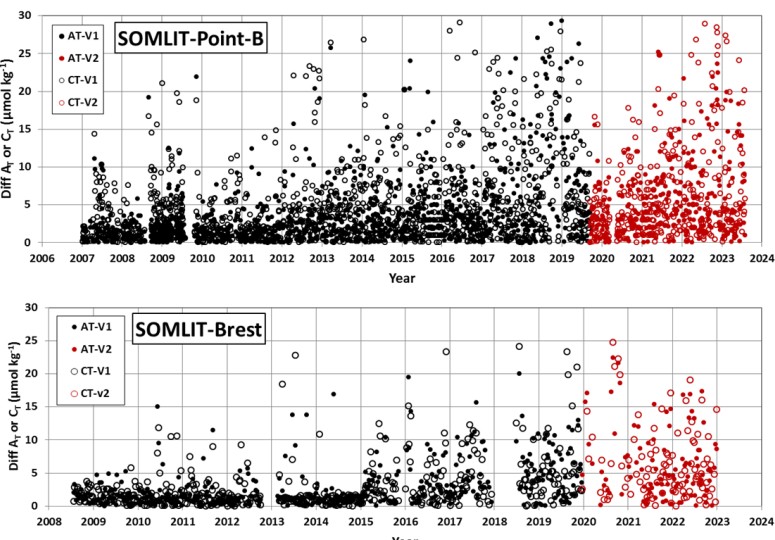

**Figure 3:** Results of duplicate $A_T$ and $C_T$ analyses from the time-series SOMLIT-Point-B in the coastal Mediterranean Sea and SOMLIT-Brest in the coastal Brittany for the data in the SNAPO-CO2-v1 dataset (black) and new data added in SNAPO-CO2-v2 (red). The plots show differences in duplicates for both $A_T$ (filled circles) and $C_T$ (open circles). Standard-deviations of these duplicates are listed in Table 2.



**Table 2:** Repeatability of $A_T$ and $C_T$ analyses for cruises with duplicate analysis. The results are expressed as the standard-deviations (Std) of the analysis of replicated samples. Nb = the number of replicates for each Time-series or Cruise. For the OISO cruises the mean repeatability was obtained from measurements at the same location (when the ship stopped).

| Cruise | Period | Nb | Std $A_T$ µmol kg$^{-1}$ | Std $C_T$ µmol kg$^{-1}$ | Reference |
|---|---|---|---|---|---|
| STEP | 2017 | 3 | 0.7 | 2.8 | Unpublished |
| CARBODISS | 2018 | 10 | 6.72 | 5.71 | Unpublished |
| SOMLIT-Point-B | 2007-2019 | 1130 | 4.5 | 5.1 | SNAPO-CO2-v1 |
| SOMLIT-Point-B | 2019-2023 | 321 | 5.2 | 6.2 | SNAPO-CO2-v2 |
| SOMLIT-Brest | 2008-2018 | 404 | 3.1 | 3.4 | SNAPO-CO2-v1 |
| SOMLIT-Brest | 2019-2022 | 142 | 6.0 | 6.1 | SNAPO-CO2-v2 |
| OISO 29 | 2019 | 46 | 1.8 | 1.8 | Leseurre et al (2022), (b) |
| OISO 30 | 2020 | 67 | 1.5 | 2.0 | Metzl et al. (2022), (b) |
| OISO 31 | 2021 | 343 | 2.6 | 3.3 | Metzl et al (2024c), (b) |

(a) See Figure 3 for the results of regular duplicates for time-series SOMLIT-Point-B, SOMLIT-Brest.
(b) Metadata and data available at www.nodc.noaa.gov/ocads/oceans/VOS_Program/OISO.html

### 3.3 Assigned flags for quality control

Identifying each data with an appropriate flag is very convenient for selecting the data (good, questionable or bad). Here we used 4 flags for each property (flags 2 = good, 3= questionable, 4=bad, and 9= no data) following the WOCE program and used in other data products such as SOCAT (Bakker et al., 2016) or GLODAP (Olsen et al., 2016; Lauvset et al., 2024). During the data-processing, we first assigned a flag for each $A_T$ and $C_T$ data based on the standard error in the calculation of $A_T$ and $C_T$ concentrations (non-linear regression, Dickson et al. 2007). By default, if the standard deviation on the regression is > 1 µmol kg$^{-1}$, we assigned a flag 3 (questionable) although the data could be acceptable and then used for interpretations. Flag 3 was also assigned when salinity was doubtful or when differences of duplicates were large (e.g. ±20 µmol kg$^{-1}$). Flags 4 (bad or certainly bad) were assigned when clear anomalies were detected for unknown reasons (e.g. a sample probably not fixed with $HgCl_2$ or analysis performed late during the COVID issue). A secondary quality control was performed by the PIs of each project based on data inspection, duplicates, $A_T$/Salinity relationship, or the mean observations in deep layers where large variability in $A_T$ and $C_T$ is unlikely to occur from year to year.

An example for quality flag is presented for all data from the MINERVE cruises conducted in 2002-2018 in the Southern Ocean where clear outliers have been identified (Figure S3). For the MINERVE cruises in 2002-2018 and a total of 12335 $A_T$ and $C_T$ analyses, 24 were identified as bad (flag 4), 978 for $A_T$ and 971 for $C_T$ listed as questionable (flag 3), and all others are considered as good data (flag 2, i.e. about 92%). For the MOOSE-GE cruises in 2021, 2022 and 2023 (new data in SNAPO-CO2-v2) and a total of 1373 $A_T$ and $C_T$ analyses, 2 were identified flagged as bad (flag 4), 38 for $A_T$ and 33 for $C_T$ listed as questionable (flag 3) all others were considered as good data (flag 2, i.e. 97%). This is better than the statistics we evaluated for the SNAPO-CO2-v1 dataset (90% flag 2 for MOOSE-GE in 2010-2019). A similar control was performed for each project.



### 3.4 Inter-comparisons

Inter-comparisons of measurements performed for different cruises or with different techniques help to evaluate the quality of the data and detect potential biases when merging the data in the same region obtained by different laboratories at different periods. This is especially important to interpret long-term trends of $A_T$ and $C_T$ as well as for $p$CO2 and pH calculated with $A_T$ $C_T$ pairs. The synthesis of various cruises in the same region and periods also offers verification and secondary control of the data.

### 3.4.1 Comparisons in deep layers

Comparisons of data in the deep layers from different cruises are useful for secondary quality control as one expects low natural variability or anthropogenic signals from season to season and over a few years. Several cruises were conducted in the Mediterranean Sea in 2017-2023 (MOOSE-GE, ANTARES and DYFAMED). The mean values of $C_T$ and $A_T$ in the deep layers (> 1800m) for each cruise confirmed the coherence of the data (Table 4). The $C_T$ and $A_T$ concentrations are also in the range of the mean values evaluated for cruises conducted in 2014 in the Mediterranean Sea (results listed in the SNAPO-CO2-v1 synthesis, Metzl et al, 2024a). In the western tropical Pacific we also observed coherent properties for the TONGA and OUTPACE cruises (Wagener et al, 2018) for data selected at 1800-2300m layer corresponding to the $C_T$ maximum layer in the Pacific Deep Water (PDW). On the other hand in the western tropical Atlantic near the Amazon River plume where the spatial variability of the properties is large at the surface (Ternon et al, 2000; Mu et al, 2021; Olivier et al, 2022) the comparison in the water column is less clear (Figure S4). Nevertheless for the AMAZOMIX and the TARA-Microbiome cruises, both conducted in September 2021, the results at close stations (around 5°N/50°W) suggest very similar concentrations at 1000m (Table 4). The comparisons in deep waters enabled to merge the different datasets for interpretations of the temporal trends and processes driving the $CO_2$ cycle in these regions (e. g. Ulses et al., 2023 and Wimart-Rousseau et al., 2023 for the Mediterranean Sea)

### 3.4.2 Comparing on board and on shore results

In surface waters where the variability is high inter-comparison is not relevant for secondary quality control. However, during the MINERVE cruises, discrete samples were occasionally performed along with sea surface underway measurements. Thus, we can compare $A_T$ and $C_T$ measured in the laboratory with those measured onboard as described by Laika et al. (2009) for the MINERVE cruises in 2005-2006. It should be noticed that the discrete samples were measured after a long trip (shipping boxes from Hobart, Tasmania to Paris, France) and thus generally analyzed at least 3 months after the cruises (cruises conducted in October to February, analyses performed in May-June). Given all the uncertainties associated to the sampling, samples storage and transport, analyses and CRMs, the mean differences between discrete and underway data are still reasonable (Std ranging between 4 and 12 µmol kg$^{-1}$, Table 5). For unknown reasons the mean difference was high for a cruise in 2008-2009 (Std > 10 µmol kg$^{-1}$, the "weather goal", Newton et al., 2015). With this in mind, we believe the MINERVE data (both underway and discrete data) are useful to interpret the change of properties in this region at seasonal or decadal scales (Laika et al., 2009; Brandon et al., 2022).



**Table 4:** Mean observations in the deep layers (> 1800m) of the Ligurian Sea (Western Mediterranean Sea for different cruises conducted in 2017-2023), of the Tropical Pacific (around 2000m for cruises in 2017 and 2019), and of the Tropical Atlantic (around 1000m for cruises in 2021). $N\text{-}A_T$ and $N\text{-}C_T$ are $A_T$ and $C_T$ normalized at salinity (S = 38 in the Ligurian Sea; S= 35 for the Pacific and the Atlantic Oceans). Nb = number of data (with flag 2). Standard deviations are in brackets.

| Cruise | Period | Nb | Pot. Temp (°C) | Salinity | N-$A_T$ (µmol kg⁻¹) | N-$C_T$ (µmol kg⁻¹) |
|---|---|---|---|---|---|---|
| **Ligurian Sea (> 1800m)** | | | | | | |
| All Cruises | 2017-2023 | 227 | 12.923 (0.052) | 38.484 (0.003) | 2558.3 (10.5) | 2300.0 (10.7) |
| DYFAMED | 2017-2022 | 74 | 12.913 (0.006) | 38.485 (0.002) | 2555.1 (11.8) | 2297.3 (12.4) |
| ANTARES | 2017-2023 | 62 | 12.944 (0.096) | 38.485 (0.005) | 2559.8 (9.0) | 2302.2 (8.9) |
| MOOSE-GE | 2017-2023 | 91 | 12.917 (0.005) | 38.484 (0.003) | 2559.8 (9.8) | 2300.7 (10.0) |
| **Tropical Pacific (layer 1800-2300m)** | | | | | | |
| OUTPACE | 2017 | 15 | 2.124 (0.055) | 34.633 (0.006) | 2414.1 (8.0) | 2318.8 (5.8) |
| TONGA | 2019 | 7 | 2.196 (0.197) | 34.619 (0.016) | 2408.9 (9.1) | 2327.2 (7.5) |
| **Western Tropical Atlantic (1000m)** | | | | | | |
| AMAZOMIX | 2021 | 14 | 4.770 (0.105) | 34.711 (0.041) | 2315.6 (20.2) | 2220.8 (17.1) |
| TARA-MICRO | 2021 | 1 | 4.852 | 34.717 | 2312.9 | 2231.1 |

**Table 5:** Comparison of $A_T$ and $C_T$ analysed on-board and at SNAPO-CO2 facilities for the MINERVE project. The results are expressed as the standard deviations (Std) of the differences for each cruise. Nb = the number of co-located samples.

| Period | Nb | Std $A_T$ µmol kg⁻¹ | Std $C_T$ µmol kg⁻¹ |
|---|---|---|---|
| 2004-2005 | 109 | 12.85 | 4.99 |
| 2005-2006 | 45 | 4.20 | 6.77 |
| 2007-2008 | 17 | 10.15 | 10.62 |
| 2008-2009 | 26 | 15.80 | 12.02 |
| 2009-2010 | 22 | 4.04 | 5.78 |
| 2010-2011 | 33 | 9.36 | 6.83 |
| 2012-2013 | 29 | 5.43 | 9.73 |

### 3.4.3 Comparison based on different techniques

Another example of comparison is presented for samples obtained in the lagoon of Mayotte Island in the western Indian Ocean and measured using different techniques. In the frame of the CARBODISS project seawater was sampled in 2018-2023 at several coral reef sites within the north-eastern part of the lagoon and measured either at LOCEAN laboratory or at La Réunion University. To remove coral sand particles the water samples were immediately filtered through Whatman GF/F filters and poisoned with mercuric chloride, following Dickson et al. (2007). In 2021, 2022 and 2023, $A_T$ was measured at La Réunion University using an automated potentiometric titration (905 Titrando Metrohm titrator with combined pH electrode 6.0253.00) and calculated from the second inflection point of the titration curve. The HCl concentration was checked each day of measurements using a CRM provided by A. Dickson, Scripps Institution of Oceanography. The $A_T$ precision based on triplicate was estimated ± 2 µmol kg$^{-1}$ (Lagoutte et al., 2023). In the studied coral reef sites $A_T$ concentrations ranged between 2250 and 2350 µmol kg$^{-1}$ but with occasional higher concentrations up to 2450-2500 µmol kg$^{-1}$. Such high $A_T$ has been observed in other coral reefs ecosystems (Cyronak et al., 2013 at Cook Island; Palacio-Castro et al., 2023 at Middle Keys, Florida). The data obtained in the lagoon of Mayotte on different coral reefs could be compared with underway observations obtained offshore of Mayotte Island (OISO-11 cruises in 2004 and CLIM-EPARSES cruise in 2019, data available in the SNAPO-CO2-v1 dataset). In the open ocean the $A_T$ concentrations ranged between 2250 and 2330 µmol kg$^{-1}$, close to the results obtained at Mayotte reefs except for samples in November 2021 that were all collected at Cratère station (12.84°S-45.39°E) (Figure 4). At this location there was a large diurnal variation in November 2021 with $A_T$ increasing from 2322 to 2508 µmol kg$^{-1}$ (Figure S5). This is because in 2021 the samples were taken at low tide recording a volcanic signal at this site allowing recording for the first time the volcanic signal in this location ($CO_2$ resurgences). In 2018 and 2019 such high $A_T$ were not measured (Figure S5) as samples were taken at high tides allowing a certain dilution of volcanic $CO_2$ emissions in the water column. Although the samples were measured with different techniques the range of $A_T$ is coherent for both datasets (Figure 4). Therefore we added the $A_T$ data measured at La Réunion University in 2021-2023 to complete the synthesis for this location (Mayotte Island).

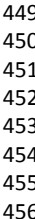
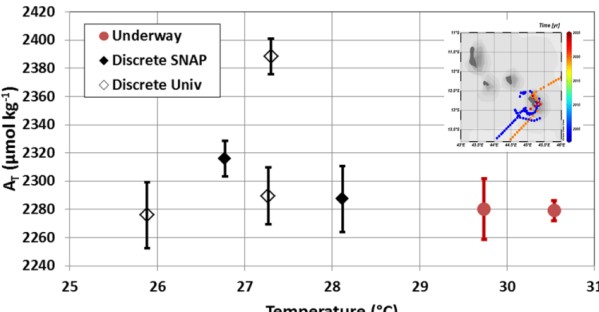

**Figure 4:** Total alkalinity ($A_T$) versus temperature for samples measured around Mayotte and in the coral reef (insert map). Underway $A_T$ was measured onboard in 2004 and 2019 (red circles) whereas discrete samples at different reef sites within the lagoon of Mayotte in 2018, 2019, 2021, 2022 and 2023 were measured at LOCEAN (black diamonds) or at La Réunion University (open diamonds). The figure presents the data averaged for each cruise in this region.





**3.4.4 Summary of quality control data**

The total number of data in the SNAPO-CO2-v2 dataset for the Global Ocean is gathered in Table 6 with corresponding flags for each property. Overall, the synthesis includes more than 91% of good data for both $A_T$ and $C_T$. About 6% are questionable and 3% are likely bad. Overall, we believe that all data (with flag 2) in this synthesis have an accuracy better than 4 µmol kg$^{-1}$ for both $A_T$ and $C_T$, the same as for quality-controlled data in GLODAP (Lauvset et al., 2024). The uncertainty ranges between the "Climate goal" (2 µmol kg$^{-1}$) and the "Weather Goal" (10 µmol kg$^{-1}$) for ocean acidification studies (Newton et al., 2015; Tilbrook et al., 2019). This accuracy is also relevant to validate or constraint data-based methods that reconstruct $A_T$ and $C_T$ fields with an error of around 10-15 µmol kg$^{-1}$ for both properties (Bittig et al., 2018; Broullón et al., 2019, 2020; Fourrier et al., 2020; Gregor and Gruber, 2021; Chau et al., 2024a).

**Table 6:** Number of Temperature, Salinity, $A_T$ and $C_T$ data in the SNAPO-CO2-v2 synthesis identified for flags 2 (good), 3 (questionable), 4 (bad), 9 (no data). Last column is the percentage of flag 2 (Good).

| Property | Flag 2 | Flag 3 | Flag 4 | Flag 9 | % flag 2 |
|---|---|---|---|---|---|
| Temperature | 68253 | 418 | 0 | 653 | 99.4 |
| Salinity | 68706 | 482 | 5 | 131 | 99.3 |
| $A_T$ | 61249 | 3910 | 2077 | 2088 | 91.1 |
| $C_T$ | 61869 | 3865 | 2057 | 1533 | 91.3 |

**4 Global $A_T$ and $C_T$ distribution based on the SNAPO-CO2-v2 dataset**

The surface distribution in the global ocean based on the SNAPO-CO2 dataset is presented in Figure 5 for $A_T$ and $C_T$. The $A_T$/Salinity and $A_T$/$C_T$ relationships are clearly identified and structured at regional scale (Figure 6). In the open ocean, high $A_T$ concentrations (> 2400 µmol kg$^{-1}$) are identified in the Atlantic subtropics (bands 35°N-15°N and 25°S-3°S) (Jiang et al., 2014; Takahashi et al., 2014). The lowest $A_T$ and $C_T$ concentrations (< 600 µmol kg$^{-1}$) are observed in the western tropical Atlantic in the Amazon River plume near the mouth (Lefèvre et al., 2017b). For $C_T$ the concentrations are high (> 2150 µmol kg$^{-1}$) in the Southern Ocean south of the polar front, associated with the deep mixing in winter and the upwelling of deep water (Metzl et al., 2006; Pardo et al., 2017). The highest $C_T$ concentrations (up to 2180-2270 µmol kg$^{-1}$) are observed in the high latitudes of the Southern Ocean near the Adélie coastal zone (MINERVE and ACE cruises), around the Kerguelen plateau (OISO-31 cruise) and close to the Antarctic Peninsula (TARA-Microbiome cruise). In the North Atlantic the new data from SURATLANT cruises in 2018-2023 confirm the high $C_T$ concentrations (> 2150 µmol kg$^{-1}$) observed in the Sub-polar gyre since 2016 due in part to the accumulation of anthropogenic $CO_2$ (Leseurre et al., 2020). Low $C_T$ concentration (< 2000 µmol kg$^{-1}$) are found in the tropics (10°N-30°S) with lower values (< 1950 µmol kg$^{-1}$) in the equatorial Atlantic band 10°N-Eq (e.g. Koffi et al., 2010; Lefèvre et al., 2021). In the Amazon shelf sector $C_T$ can reach even lower concentration (< 1700 µmol kg$^{-1}$, AMAZOMIX cruise).

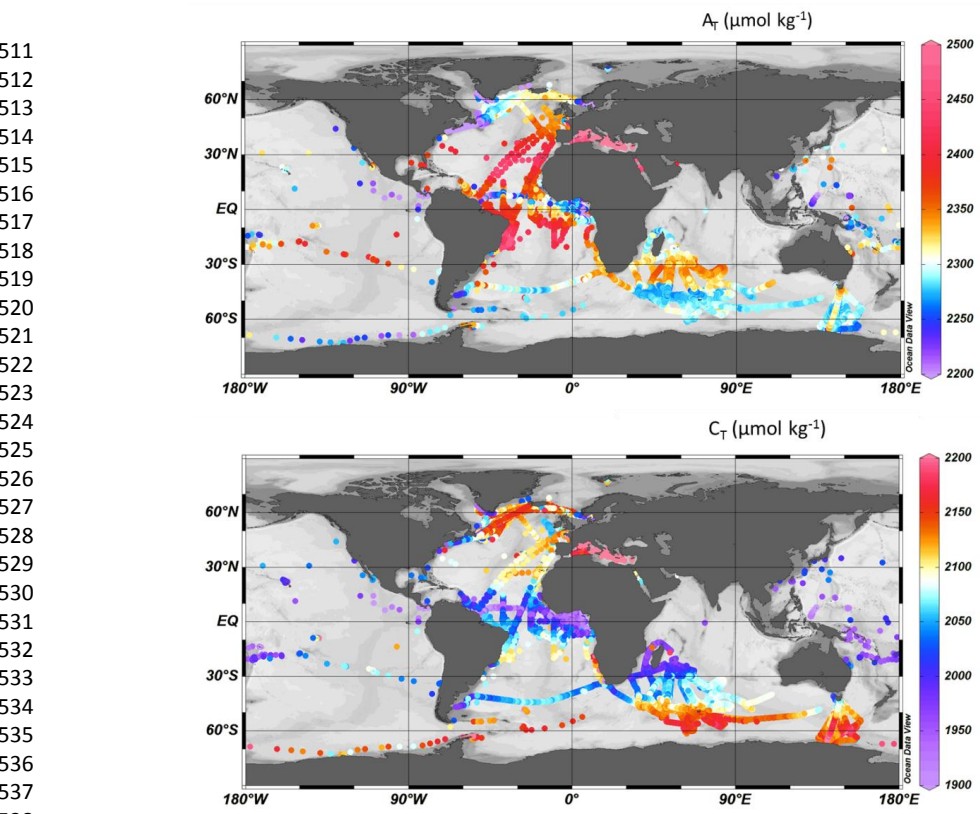

**Figure 5**: Distribution of $A_T$ (top) and $C_T$ (bottom) concentrations (µmol.kg$^{-1}$) in surface waters (0-10m) in the SNAPO-CO2-v2 dataset. Only data with flag 2 are presented in these figures. Figures produced with ODV (Schlitzer, 2018).

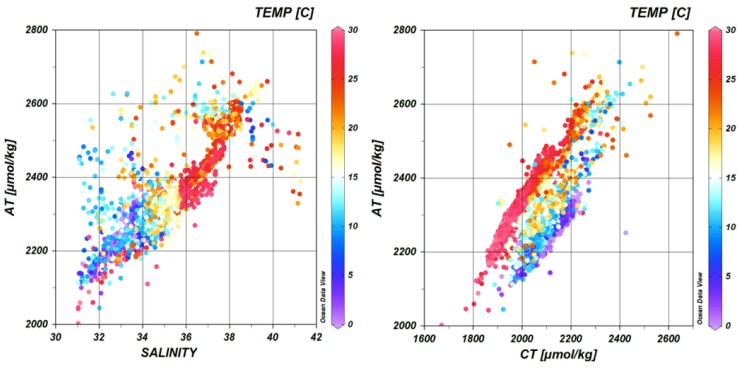

**Figure 6:** Relationships between $A_T$ and Salinity (left panel) and $A_T$ versus $C_T$ (right panel) for samples in surface waters (0-10m and Salinity > 31). Only data with flag 2 are presented (nb = 48749). The color scales correspond to the temperature. The data not aligned correspond to coastal zones (e.g. COCORICO2 stations). Figures produced with ODV (Schlitzer, 2018).





## 5 Regional $A_T$ and $C_T$ distributions and trends based on the SNAPO-CO2 dataset

### 5.1 The Mediterranean Sea

Compared to the open ocean, $A_T$ concentrations are much higher in the Mediterranean Sea (Copin-Montégut, 1993; Schneider et al., 2007; Álvarez et al., 2023) with values up to 2600 µmol kg$^{-1}$. The $A_T$ and $C_T$ data obtained in 2014-2023 show a clear contrast between the northern and southern regions of the Western Mediterranean Sea with higher concentration in the Ligurian Sea and the Gulf of Lion (Figure 7). This contrast is associated to the circulation and the frontal system in this region (e.g. Barral et al, 2021). New data in the coastal zones in the Gulf of Lion (ACCESS, DICASE, CARBODELTA, COCORICO2, MESURHOBENT) also have very high $A_T$ and $C_T$ concentrations ($A_T$ >2600 µmol kg$^{-1}$; $C_T$ >2350 µmol kg$^{-1}$). Very low $A_T$ and $C_T$ concentrations ($A_T$ <2500 µmol kg$^{-1}$; $C_T$ < 2200 µmol kg$^{-1}$) were also occasionally observed in the coastal zones (COCORICO2 stations, Petton et al, 2024).

In summer 2022 the Mediterranean Sea experienced an exceptional warming (Figure S6) superposed to the long-term warming in the ocean (Cheng et al, 2024). Such event would impact the internal ocean processes such as thermodynamic, stratification and biological processes (Coppola et al., 2023) and the inter-annual variability and trends of $C_T$, pH, $f$CO$_2$ and air-sea CO$_2$ fluxes (Yao et al., 2016; Wimart-Rousseau et al., 2023; Chau et al., 2024b). As in 2003, the warming in summer 2022 was associated to the drought event that occurred in Europe and over the Mediterranean Sea (Faranda et al., 2023). In July 2022, the maximum temperature of 28.42°C was observed at station SOMLIT-Point-B. In the Ligurian Sea the temperature trend was faster in recent years, +0.173 ± 0.072 °C per decade over 1990-2010 and +0.678 ± 0.143 per decade over 2010-2023 (Figure S6). With the new data added in the SNAPO-CO2-v2 synthesis (DYFAMED, MOOSE-ANTARES, and MOOSE-GE) we evaluated a temperature trend of +0.84 ± 0.20 °C per decade over 1998-2022 indicating that the discrete sampling captured the property changes at regional scale. Based on the data in the Ligurian Sea the trends of $C_T$ appeared faster in summer (+1.53 ± 0.46 µmol kg$^{-1}$ yr$^{-1}$) than in winter (+0.94 ± 0.64 µmol kg$^{-1}$ yr$^{-1}$, Table 7). On the other hand, the trends of $A_T$ were the same (+0.72 ±0.36 µmol kg$^{-1}$ yr$^{-1}$ in winter and +0.69 ± 0.42 µmol kg$^{-1}$ yr$^{-1}$ in summer). The trend of $C_T$ in surface in winter was close to the one derived at 100m (below the Chl-a maximum), $C_T^{100m}$ = +1.10 ±0.17 µmol kg$^{-1}$ yr$^{-1}$ (Figure 8) whereas for $A_T$ the trend was the same in surface and at depth (+0.76 ±0.12 µmol kg$^{-1}$ yr$^{-1}$). This suggests that the winter $C_T$ data recorded the anthropogenic CO$_2$ uptake of around +1 µmol kg$^{-1}$ yr$^{-1}$, Figure S7). As noted by Touratier and Goyet (2009) the $C_T$ concentrations in the Mediterranean Sea should increase in parallel with the level of atmospheric anthropogenic CO$_2$. For an atmospheric CO$_2$ rate of +2.16 ppm yr$^{-1}$ over 1998-2023 (Lan et al., 2024) and at fixed sea surface temperature (17.75°C), salinity (38.25) and $A_T$ (2567 µmol kg$^{-1}$), the theoretical $C_T$ increase would be +1.24 µmol kg$^{-1}$ yr$^{-1}$. Interestingly, an anthropogenic flux of -0.3 ±0.02 molC m$^{-2}$ yr$^{-1}$ in the Mediterranean Sea (Bourgeois et al., 2016) would correspond to an increase of $C_T$ of 1.07 ±0.07 µmol kg$^{-1}$ yr$^{-1}$ in the top 100 meters. This is again close to what is observed in winter or at 100m (Table 7, Figure 8). On the other hand the faster $C_T$ trend observed in surface waters during summer might be associated with a decrease in biological production and/or changes in circulation/mixing over time that deserve specific investigations such as analyzed for the oxygen budget in this region (Ulses et al, 2021). It is worth noting that the $C_T$ and $A_T$ trends in coastal zones of the Mediterranean Sea are opposite to those observed offshore: for example at station

SOLEMIO (Bay of Marseille, Wimart-Rousseau et al., 2020) the $C_T$ and $A_T$ concentrations decreased over 2016-
2022 and thus opposed to the anthropogenic $CO_2$ signal, indicating that processes such as riverine inputs,
advection or biology control the carbonate system decadal variability at local scale. This calls for developing
dedicated complex biogeochemical models to resolve these processes (Barré et al., 2023, 2024), especially when
extreme events occurred, such as the very hot summer in 2024 with SST up to 30°C in the Mediterranean Sea
(Platforms Buoy/Mooring AZUR, EOL and La Revellata, data available at https://dataselection.coriolis.eu.org/).
The data obtained in the Mediterranean Sea are important not only to validate biogeochemical models but also to
reconstruct the carbonate system from $A_T$ and $pCO_2$ data (Chau et al., 2024a) as the global $A_T$/SSS relationships
(e.g. Carter et al., 2018) are not suitable for this region.

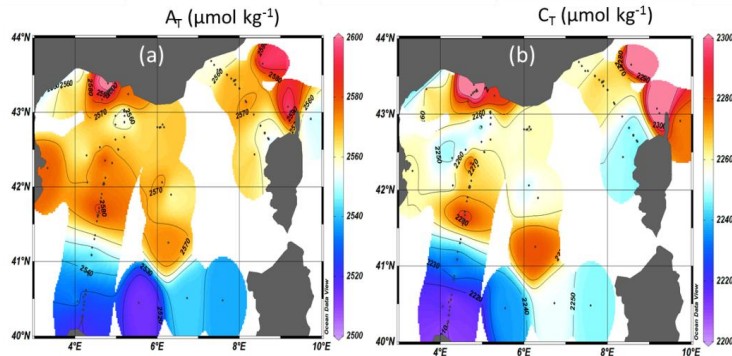

**Figure 7:** Distribution of $A_T$ (a) and $C_T$ (b) in µmol kg$^{-1}$ in surface waters of the Mediterranean Sea (0-10m) from observations over 2014-2023. Figures produced with ODV (Schlitzer, 2018).

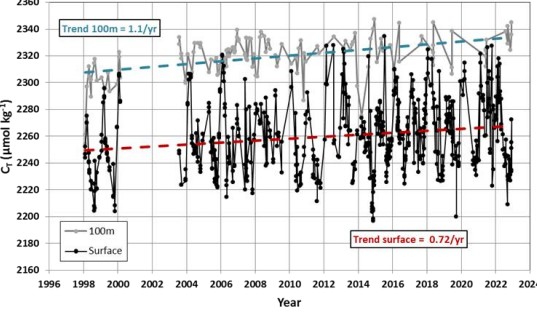

Figure 8: Time-series of $C_T$ concentrations in surface (black symbols) and at 100m (grey symbols) in the Ligurian Sea. The trends over 1998-2022 is surface (red) and at 100m (blue) are indicated by dashed lines.

**5.2 The North Atlantic**

The North Atlantic Ocean is an important $CO_2$ sink (Takahashi et al. 2009) due to biological activity
during summer, heat loss and deep convection during winter. As a result this region contains high concentrations
of anthropogenic $CO_2$ ($C_{ant}$) in the water column (Khatiwala et al., 2013). Decadal variations of the $C_{ant}$
inventories were recently identified at basin scale probably linked to the change of the overturning circulation
(Gruber et al., 2019; Müller et al., 2023; Pérez et al., 2024). This region experienced climate modes such as the
North Atlantic Oscillation (NAO) and the Atlantic Multidecadal Variability (AMV) that imprint variability in
air-sea $CO_2$ fluxes at inter-annual to multidecadal scales (e.g. Thomas et al., 2008; Jing et al., 2019;
Landschützer et al., 2019) but not always clearly revealed at regional scale (Metzl et al., 2010; Schuster et al.,
2013; Pérez et al., 2024). In addition it has been recently shown that extreme events such as the marine heat
wave in summer 2023 leaded to a reduce $CO_2$ uptake in this region (Chau et al., 2024b). Although the annual
$CO_2$ fluxes deduced from Global Ocean Biogeochemical Models (GOBM) seem coherent with the data-products
at basin scale (resp. -0.30 ±0.07 and -0.24 ±0.03 PgC/yr for the NA-SPSS biome) the $pCO_2$ cycle seasonality is
not well simulated (Pérez et al., 2024). Therefore to correct the GOBMs outputs, comparisons with the observed
$C_T$ and $A_T$ cycles are also needed.
In this context regular sampling in the North Atlantic (OVIDE cruises, Mercier et al., 2015, 2024;
SURATLANT transects, Reverdin et al., 2018) and time-series stations in the Irminger and Iceland Seas
(Ólafsson, et al., 2010; Lange et al., 2024; Yoder et al., 2024) are important to explore the variability of the
biogeochemical properties from seasonal (Figure S8) to decadal scales (Figure 9). The SURATLANT data added
in the SNAPO-CO2-v2 dataset over 2017-2023 offer new observations in the North Atlantic Subpolar Gyre
(NASPG in the NA-SPSS biome) and new transects from Norway to Iceland and reaching the coast of Greenland
(Figure 9). In 2010 the winter NAO was negative, moved to a positive state in 2012-2020 and was again very
low in 2021. The new SURATLANT data after 2017 confirm the cooling and the freshening in the NASPG since
2009 (Holliday et al., 2020; Leseurre et al., 2020; Siddiqui et al., 2024) whereas the most recent data in 2022 and
2023 suggest a reverse trend (increase of salinity and temperature, not shown). After 2016, large $C_T$ anomalies in
the NASPG were observed. For examples, in April 2019 and 2022, the $C_T$ concentrations were low compared to
2016 (Figure 9) and opposed to the expected anthropogenic $CO_2$ uptake. In September 2023 the $C_T$
concentrations were much lower than in 2022 (Figure 9) probably linked to biological productivity when the
NAO index was negative (Fröb et al., 2019) as observed in summer 2023 (NAO < -2 in July 2023). Despite these
variability the $C_T$ trends are relatively well evaluated (Table 7). As in the Mediterranean Sea the $C_T$ trends in the
NASPG appeared different depending on the season (Figure 9). The $C_T$ increase was faster in September than in
April (resp. +1.09 ± 0.37 µmol kg$^{-1}$ yr$^{-1}$ and +0.78 ± 0.23 µmol kg$^{-1}$ yr$^{-1}$). This is either close to or lower than the
theoretical $C_T$ increase due to the rising of atmospheric $CO_2$ (+0.91 µmol kg$^{-1}$ yr$^{-1}$) and in the range of recent
results evaluated for the Sub-polar Mode Waters in the Irminger Sea ($C_{ant}$ trend = 0.95 ± 0.17 µmol kg$^{-1}$ yr$^{-1}$ for
the period 2009-2019, Curbelo-Hernández et al., 2024).


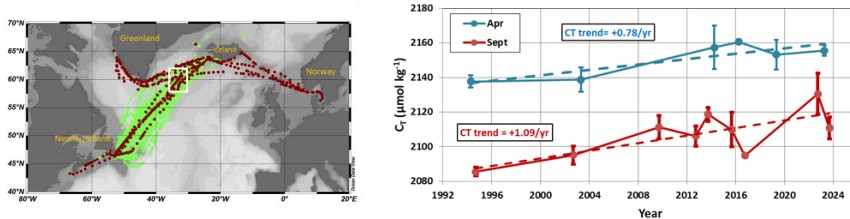

**Figure 9:** Left: Data in SNAPO-CO2-v1 (green) and new data in v2 (brown) from the SURATLANT cruises in
1993-2023 in the North Atlantic. Figure produced with ODV (Schlitzer, 2018). The white box identified the
region of selected data around 60°N for the trend analysis. Right: Time-series of average $C_T$ concentrations in
April (blue) and September (red) in this region. The trends for each season are indicated (see also Table 7).



### 5.3 The Tropical Atlantic

In the Tropical Atlantic, previous studies highlighted the large variability of biogeochemistry and the difficulty in detecting long-term trends of $C_T$ (e.g. Lefèvre et al., 2021). This is related to the variability of circulation, equatorial upwelling, biological processes (some linked to Saharan dust) and inputs from large rivers (Congo, Amazon and Orinoco). The new data added in version SNAPO-CO2-v2 (Figure S9) show the contrasting zonal $C_T$ distribution in this region with lower concentrations in low salinity regions of the North Equatorial Counter Current and Guinea Current (Figure 5; Oudot et al., 1995; Takahashi et al., 2014; Broullón et al., 2020; Bonou et al., 2022). For exploring the temporal changes we selected the data in the western region available for at least 10 years and separated the northern and southern sectors. In both regions the $C_T$ trend is close to +3 µmol kg$^{-1}$ yr$^{-1}$ (Table 7, Figure S9) much higher than the excepted anthropogenic signal. In this region where coastal water masses mixes with oceanic waters, the inter-annual variability of $C_T$ is large and the changes driven by competitive processes (circulation, biological processes). More observations and dedicated models are needed to separate the anthropogenic and natural variability in this region (Pérez et al., 2024).

### 5.4 The Southern Ocean

In the Southern Ocean there are a few regular multi-annual observations of the carbonate system. Time series of more than 10 years were obtained in the Drake Passage (Munro et al., 2015) and in the Southern Indian Ocean (Leseurre et al., 2022; Metzl et al., 2024b). Observations were also obtained for more than 20 years southeast of New Zealand at the Munida Time Series (MTS) in the subtropical and sub-Antarctic frontal zones (Currie et al., 2011; Vance et al., 2024). To complement these datasets we have added the data collected in the South-Eastern Indian Ocean between Tasmania and Antarctica in the frame of the MINERVE cruises (Figure 10; Brandon et al., 2022). These cruises were conducted from October to March offering each year a view of the seasonal changes between late winter and summer from the sub-Antarctic zone to the coastal zone near Antarctica (Adélie land). In all sectors (here from 45°S to 67°S) the $C_T$ concentrations were higher in October when the mixed-layer depth (MLD) was deep and were lower during the productive summer season (e.g. Laika et al., 2009; Shadwick et al., 2015). An example is presented at 60°S/151°E from the data obtained along a reoccupied track in 2011-2012 (Figure S10). At this location south of the Polar Front in the POOZ/HNLC area, the $C_T$ concentrations were +25 µmol kg$^{-1}$ higher in October compared to February. The same seasonal amplitude was observed in the western Indian sector of the POOZ (Metzl et al., 2006, 2024b) suggesting that the $C_T$ seasonality is relatively homogeneous in this region corresponding to the Indian SO-SPSS biome (Fay and McKinley, 2014). The difference in the climatological $C_T$ between October and January is on average +28.3 ± 9.8 µmol kg$^{-1}$ in the Indian Ocean POOZ (Takahashi et al., 2014). Given this seasonality and potential change in the seasonal amplitude over time (Gallego et al., 2018; Landschützer et al., 2018; Shadwick et al., 2023) the property trends have to be evaluated for October and January-February separately, here over 2002-2012 in the POOZ (Figure 10, Table 7). In both seasons, the average $C_T$ concentrations reached a minimum in 2008 and increased faster in 2008-2012 (up to +4.8 µmol kg$^{-1}$ yr$^{-1}$). Interestingly, such acceleration of the trend after 2009 was observed for $pCO_2$ at the MTS station (Vance et al., 2024). We note that the $C_T$ trend over 2002-2012 was



slightly faster in October (Figure 10) probably linked to deeper MLD as suggested from the cooling and the
salinity increase observed during this season (not shown).












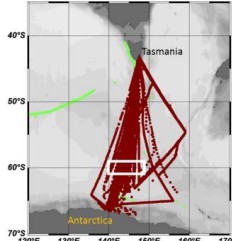 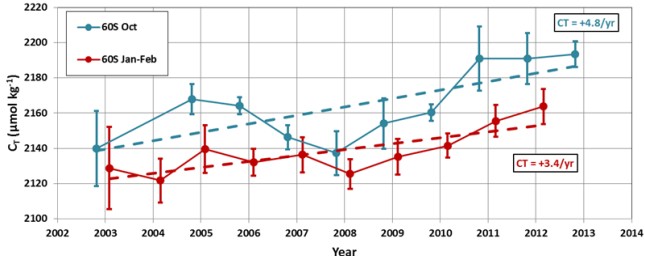

**Figure 10:** Left: Data in SNAPO-CO2-v1 dataset (green) and new data in version v2 (brown) in the South
eastern Indian Ocean. Figure produced with ODV (Schlitzer, 2018). The white box identified the region of
selected data around 60°S for the trend analysis. Right: Time-series of average $C_T$ concentrations in January-
February (red) and October (blue) around 60°S (white box in the map). The trends for each season are indicated
(see also Table 7).

In the western Indian sector, the new data in the SNAPO-CO2-v2 dataset from the OISO cruises at high
latitudes also recorded a rapid $C_T$ trend over 5-8 years periods (e.g., +3.4 µmol kg$^{-1}$ yr$^{-1}$ in 2015-2020 at 56°S,
Figure 11, Table 7). Although the inter-annual variability of $C_T$, between 10 and 20 µmol kg$^{-1}$, is often
recognized (Figure 11), the evaluation of the trends over more than 20 years indicated faster trend in the
subtropical Indian Ocean (+1.1 µmol kg$^{-1}$ yr$^{-1}$) compared to higher latitudes (Indian POOZ, +0.6 µmol kg$^{-1}$ yr$^{-1}$);
they are close to the expected anthropogenic signal in these regions (+1.1 µmol kg$^{-1}$ yr$^{-1}$ in the subtropics and
+0.8 µmol kg$^{-1}$ yr$^{-1}$ at higher latitudes).















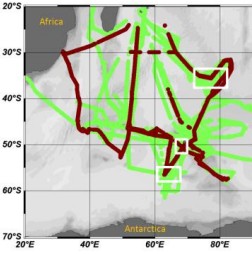 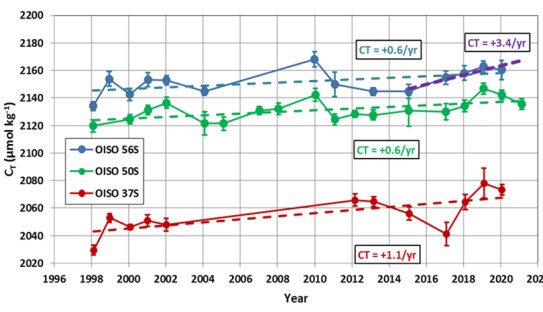

**Figure 11:** Left: Data in SNAPO-CO2-v1 dataset (green) and new data in version v2 (brown) in the South
Western Indian Ocean (OISO cruises). Figure produced with ODV (Schlitzer, 2018). The white boxes identified
the regions of data selected around 37°S, 50°S and 56°S for the trend analysis. Right: Time-series of average $C_T$
concentrations in January-February at 37°S (red), 50°S (green) and 56°S (blue). The trends for each region are
indicated (see also Table 7).






**Table 7:** Trend of $C_T$ (µmol kg$^{-1}$ yr$^{-1}$) and corresponding standard error in selected regions where data are available for more than 10 years. The projects/cruises for selection of the data in each domain are indicated.

| Region | Period/Season | $C_T$ trend (µmol kg$^{-1}$ yr$^{-1}$) | Projects/Cruises |
|---|---|---|---|
| North Atlantic (NASPG) | 1994-2023 April | +0.78 (0.23) | SURATLANT |
| North Atlantic (NASPG) | 1994-2023 September | +1.09 (0.37) | SURATLANT |
| West. Trop. Atl. 5N-Eq | 2009-2021 April-October | +3.31 (2.13) | AMAZOMIX, PIRATA-BR, TARA |
| West. Trop. Atl. Eq-10S | 2005-2015 April-October | +3.05 (1.64) | CAMFIN-WAT, PIRATA-BR, VOS |
| Ligurian Sea 8E | 1998-2022 Jan.-Feb. | +0.94 (0.64) | ANTARES, DYFAMED, MOOSE-GE |
| Ligurian Sea 8E | 1998-2023 July-August | +1.53 (0.46) | ANTARES, DYFAMED, MOOSE-GE |
| Subtropical Indian 37S | 1998-2020 Jan.-Feb. | +1.12 (0.36) | OISO |
| South West. Indian 50S | 1998-2021 Jan.-Feb. | +0.61 (0.21) | OISO |
| South West. Indian 56S | 1998-2020 Jan.-Feb. | +0.58 (0.27) | OISO |
| South West. Indian 56S | 2015-2020 Jan.-Feb. | +3.41 (0.73) | OISO |
| South East. Indian 60S | 2002-2012 Jan.-Feb. | +3.37 (0.94) | MINERVE, OISO |
| South East. Indian 60S | 2002-2012 October | +4.79 (1.62) | MINERVE |

## 5.5 The Coastal Zones

Coastal waters experience enhanced ocean acidification due to increasing $CO_2$ uptake, accumulation of anthropogenic $CO_2$ (Bourgeois et al 2016; Laruelle et al, 2018; Roobaert et al, 2024a; Li et al, 2024) and from local anthropogenic inputs through rivers or from air pollution (e.g. Sarma et al, 2015; Sridvi and Sarma, 2021; Wimart-Rousseau et al, 2020). The changes of the $CO_2$ uptake in coastal zones are also linked to biological processes (Mathis et al, 2024) or to circulation and local upwelling (Roobaert et al, 2024b), all controlling large variability of $A_T$ and $C_T$ in space and time leading to uncertainties for detecting long-term changes of $pCO_2$ and air-sea $CO_2$ fluxes in heterogeneous coastal waters (Dai et al 2022; Resplandy et al, 2024). At seasonal scale, large differences between observations and models were also identified leading to differences in the coastal ocean $CO_2$ sink up to 60% (Resplandy et al, 2024). It is thus important to document the seasonal cycles of $A_T$ and $C_T$ to compare and correct models and thus to better predict future changes of biogeochemical properties in coastal waters and their impact on marine ecosystems. A better understanding of the processes and their retroaction in the coastal regions is also required regarding Marine Carbone Dioxide Removal (MCDR) experiments and for their evaluation (e.g. Ho et al, 2023).

In the SNAPO-CO2-v2 dataset new data have been added in the coastal zones at stations SOMLIT-Brest, SOMLIT-Roscoff and SOMLIT-Point-B. They extend the period to 2022 or 2023 for temporal analysis. New data in the French coastal zones have been also included from the COCORICO2 project documented in detail by Petton et al (2024). The observations in coastal zones could be identified in the MARCATS regions (Margins and CATchment Segmentation, Laruelle et al, 2013) (Figure 12) where little information is available for quantifying the ocean $CO_2$ sink at the decadal scale and for evaluation of the anthropogenic $CO_2$ uptake (Regnier et al, 2013; Dai et al, 2022; Li et al, 2024). To explore the change of the observed properties in the coastal zones and have a flavor of the long-term $C_T$ trends we selected the time series with at least 10 years of data (Table 8, Figure 13). Except at high latitudes (Greenland and Antarctic coastal zones), we observed a warming in coastal zones (not shown). Changes in salinity are also identified (increase or decrease) and results

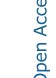

of the trends are presented for salinity-normalized $C_T$ at 34, 35 or 38 depending on the region. Although the
inter-annual variability is large in coastal waters, sometimes linked to extreme events (e.g. river discharges), we
observed an increase in N-$C_T$ at most of the 8 selected locations. The exceptions are the coastal zones in the Gulf
of Lion near the Rhone River and near Tasmania in October.
In the Gulf of Lion, the new data in the coastal zone confirmed the first view at the SOLEMIO station
over 2016-2018 (Bay of Marseille, Wimart-Rousseau et al, 2020). In this region the lowest $C_T$ was observed in
summer 2022 (average $C_T$ of 2238.6 ±21.0 µmol kg$^{-1}$), much lower than in 2015 (2290.8 ±44.7 µmol kg$^{-1}$). Over
the continental shelf south of Tasmania (MARCATS #34), the trend in N-$C_T$ was positive in summer but not
significant in October. In October this was associated with an increase in Salinity and in $A_T$ probably linked to
advective processes via the reversal and variability of the Zeehan or the East Australian currents. From our data a
warming of +0.06°C yr$^{-1}$ was identified for both seasons over 2002-2012 as previously observed south of
Tasmania over 1991-2003 impacting the $p$CO2 trend and air-sea $CO_2$ fluxes in this region (Borges et al, 2008).
The difference in the N-$C_T$ trends in austral summer and spring calls for new detail studies with extended data in
this region. At high latitude in the Adélie Land (Antarctic coast MARCATS #45), the variability of N-$C_T$ was
large (range from 2150 to 2200 µmol kg$^{-1}$, Figure 13) and the trend over 10 years in summer was not significant
(Table 8). As opposed to the open zone at 60°S (Figure 10) the $C_T$ concentrations in the coastal zone near
Antarctica were not increasing, probably linked to competitive processes between anthropogenic uptake, changes
in primary production, mixing or ice melting (Shadwick et al, 2013, 2014). More data are needed to better
evaluate the changes of the carbonate system in Antarctic coastal zones where bottom waters are formed and
transport anthropogenic $CO_2$ at lower latitudes (Zhang et al, 2023).
For the coastal time series SOMLIT where annual trends could be estimated (sampling at monthly
resolution), the N-$C_T$ increase (+2.1 to 3.4 µmol kg$^{-1}$ yr$^{-1}$) is close or higher than the anthropogenic signal leading
to a decrease in pH ranging between -0.05 to -0.06 TS decade$^{-1}$. The new data added in the SNAPO-CO2-v2
dataset (2016-2023) confirm the progressive increase in $C_T$ and the acidification in the western Mediterranean
Sea and in the North-East Atlantic coastal zones (Kapsenberg et al, 2017; Gac et al, 2021).

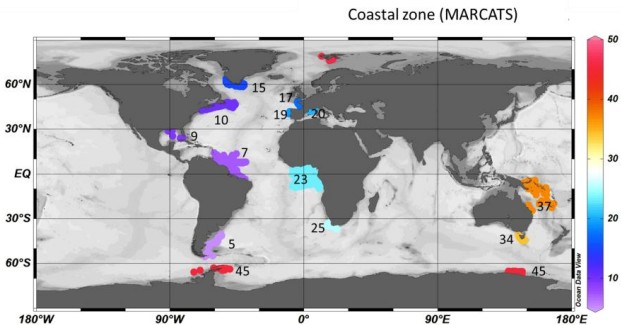

**Figure 12:** Location of $A_T$ $C_T$ data available in the coastal zones in the SNAPO-CO2-v2 dataset. Numbers and Color code identify MARCATS region (Laruelle et al, 2013). Figure produced with ODV (Schlitzer, 2018).



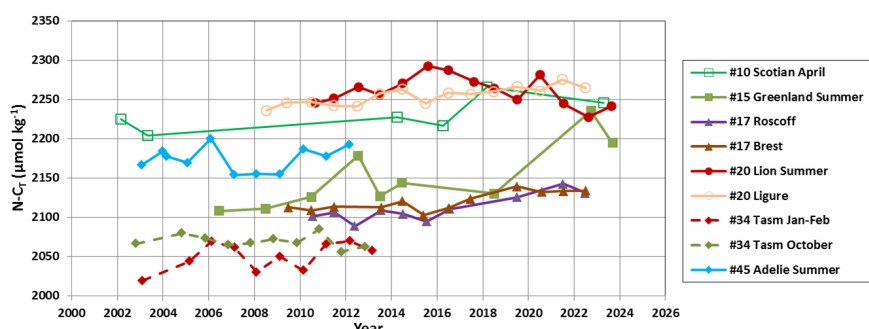

**Figure 13:** Time-series of average N-$C_T$ concentrations (µmol kg$^{-1}$) in selected MARCATS regions for different period when data are available for ten years or more. The trends and periods for each region are indicated in Table 8.

**Table 8:** Trends of N-$C_T$ (µmol kg$^{-1}$ yr$^{-1}$) and corresponding standard errors in selected coastal regions where data are available for 10 years or more. The projects/cruises for selection of the data in each domain are indicated. MARCATS # regions also identified. Salinity value used for $C_T$ normalization indicated.

| Region #MARCATS | Period | Season | N-$C_T$ Ttrend (µmol kg$^{-1}$ yr$^{-1}$) | Salinity | Projects/Cruises |
|---|---|---|---|---|---|
| Scotian #10 | 2002-2023 | March-April | +1.71 (0.97) | 35 | SURATLANT |
| Greenland #15 | 2006-2023 | June-mid-Sept | +5.77 (1.62) | 35 | OVIDE, SURATLANT |
| Roscoff #17 | 2010-2022 | All season | +3.40 (0.76) | 35 | CHANNEL, COCORICO2, SOMLIT ROSCOFF |
| Bay of Brest #17 | 2009-2022 | All seasons | +2.17 (0.52) | 35 | SOMLIT-Brest, COCORICO2, ECOSCOPA, |
| LION#20 | 2010-2023 | June-Sept | -1.19 (1.25) | 38 | COCORICO2, MOOSE-GE, SOLEMIO (a) |
| LIGURE#20 | 2008-2022 | All seasons | +2.12 (0.36) | 38 | SOMLIT-Point-B, MOOSE-GE |
| Tasmania #34 | 2003-2013 | Jan-Feb | +2.73 (1.72) | 35 | MINERVE, OISO |
| Tasmania #34 | 2002-2012 | Oct | -0.65 (0.89) | 35 | MINERVE, OISO |
| Adélie #45 | 2002-2012 | Dec-Feb | +0.63 (0.70) | 34 | MINERVE, OISO |

(a)     For LION, some data in summer were also used from punctual cruises: AMOR-BFlux, CARBORHONE, DICASE, LATEX, MESURHOBENT, MISSRHODIA2 and MOLA.

**6 Summary and suggestions**

This work extends in time and new oceanic regions the $A_T$ and $C_T$ data presented in the first SNAPO-CO2 synthesis (Metzl et al, 2024a). It includes now more than 67 000 surface and water column observations in all oceanic basins, in the Mediterranean Sea, in the coastal zones, near coral reefs, and in rivers. The data synthesized in version v2 are based on measurements of $A_T$ and $C_T$ performed between 1993 and 2023 with an accuracy of ±4 µmol kg$^{-1}$. Based on a secondary quality control, 91% of the $A_T$ and $C_T$ data are considered as good (WOCE Flag 2) and 6% probably good (Flag 3). For the open ocean this synthesis complements the SOCAT, GLODAP and SPOTS data products (Bakker et al., 2016; Lauvset et al., 2024; Lange et al, 2024). For the coastal sites this also complements the synthesis of coastal time-series in the Iberian Peninsula (Padin et al, 2020), in the Canadian Atlantic continental shelf (Gibb et al, 2023) and around North America (Fassbender et al., 2018; Jiang et al., 2021; Jiang et al 2024, in prep). The SNAPO-CO2 dataset enables to investigate the seasonal cycles, the inter-annual variability and the decadal trends of $A_T$ and $C_T$ in various oceanic provinces. The same

temporal analyses could be investigated for other carbonate system properties such as $f\text{CO}_2$ or pH calculated
from $A_\text{T}$ and $C_\text{T}$ for air-sea $\text{CO}_2$ flux estimates or ocean acidification studies (Figure 14).

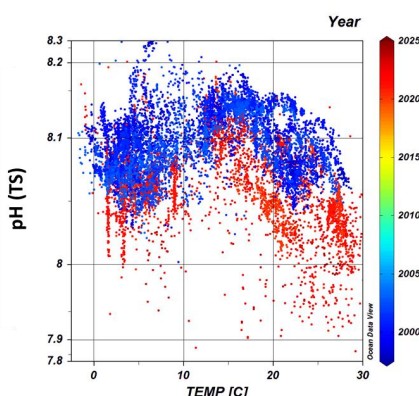

**Figure 14:** An example of observed ocean acidification derived from the SNAPO-CO2-v2 dataset: pH (TS)
calculated with $A_\text{T}$ and $C_\text{T}$ data are presented as a function of temperature (°C) for years 1998-2002 (blue
symbols) and 2020-2023 (red symbols) and for salinity > 33 (Nb data selected with flag 2 = 11994). In recent
years the pH was lower. Figure produced with ODV (Schlitzer, 2018).

In almost all regions the new data in 2021-2023 indicated that the $C_\text{T}$ concentrations were higher in
recent years. In regions where data are available for more than 2 decades, the time-series show an increase of sea
surface $C_\text{T}$ (North Atlantic, Southern Indian Ocean and Ligurian Sea) with a rate close to or higher than the
changes expected from anthropogenic $\text{CO}_2$ uptake. It is also recognized that at seasonal scale the $C_\text{T}$ trends could
be different. However, with the data in hand, the long-term trend of $C_\text{T}$ cannot be quantified with confidence to
compare with the anthropogenic carbon uptake in some regions. This is the case in the eastern tropical Atlantic
subject to high inter-annual variability (Lefèvre et al., 2021, 2024) although new data have been added over
2005-2022 in this region (Table 1, Figure S9). When data are available for less than a decade the increase in $C_\text{T}$
was observed but the trend was uncertain due to large inter-annual variability (e.g. Adélie Land). An exception
was identified in the coastal zone in the Gulf of Lion (Mediterranean Sea) where summer data since 2010 present
a decrease in $C_\text{T}$ most pronounced since 2015 ($C_\text{T}$ trend = -5.2 ± 1.5 µmol kg$^{-1}$ yr$^{-1}$). Such $C_\text{T}$ decrease over 10
years was also observed at the Hawaii Ocean Time series, HOT over 2010-2020 (Dore et al, 2009,
https://hahana.soest.hawaii.edu/hot/hotco2/hotco2.html, last access: 27 August 2024).
Although the $A_\text{T}$ concentrations present significant inter-annual variability such as in the NASPG, in the
Topical Atlantic or the Adélie land and coastal zones, $A_\text{T}$ appears relatively constant over time except at these
locations. In the open ocean, we observed an increase of $A_\text{T}$ in the Southern Ocean south of the Polar Front
around 60°S in 2003-2012 not directly linked to salinity. In the coastal zones a decrease of $A_\text{T}$ was pronounced
south of Greenland. In the coast in the Gulf of Lion, as observed for $C_\text{T}$, $A_\text{T}$ decreased ($A_\text{T}$ trend = -2.8 ±1.2 µmol
kg$^{-1}$ yr$^{-1}$). This is opposed to the changes observed in the Ligurian Sea at station SOMLIT-Point-B, where $C_\text{T}$ and
$A_\text{T}$ increased over 2007-2015 (Kapsenberg et al, 2017) highlighting the contrasting $C_\text{T}$ and $A_\text{T}$ trends in the
Mediterranean coastal zones where ocean acidification is detected (here over 2008-2022, pH trend of -0.048
±0.003.decade$^{-1}$). With the continuous warming, reduced stratification and the rapid pH change observed in the
Mediterranean Sea, how the marine ecosystems will respond in the future should be addressed (e.g. Howes et al,



2015; Maugendre et al 2015; Lacoue-Labarthe et al, 2016). The SNAPO-CO2-v2 dataset could also be used to
explore and analyze the changes of the carbonate system occurring during extreme events such as marine heat
waves, rapid freshening, deep convection or high phytoplankton bloom events.
This dataset would also serve for validating autonomous platforms capable of measuring pH and $fCO_2$
properties (Sarmiento et al, 2023) and, along with other synthesis products (Jiang et al, 2024 in prep.), provides
an additional reference dataset for the development and validation of regional biogeochemical models for
simulating air-sea $CO_2$ fluxes. Thanks to the RECCAP2 stories, it has been recognized that Ocean
Biogeochemical Models present biases in the seasonal cycle of $C_T$ and $A_T$ due to inadequate representation of
biogeochemical cycles (e.g. Hauck et al, 2023; Rodgers et al, 2023; Sarma et al, 2023; Pérez et al, 2024;
Resplandy et al, 2024). The SNAPO-CO2-v2 dataset could be used to guide analyses for regional or global
biogeochemical models for $A_T$ and $C_T$ comparison and validation from seasonal to decadal scales. Our dataset is
also essential for training and validating neural networks capable of predicting variables in the carbonate system
(e.g. Fourier et al, 2020; Chau et al, 2024a; Gregor et al, 2024), thereby enhancing observations of marine $CO_2$ at
different spatial and temporal scales. Furthermore, we encourage the use of this dataset (or part of it), at sea or
prior going to sea for cruise planning. Indeed, using the approach of Davis and Goyet (2021) which takes into
account the multiple constraints (ship-time, number of samples, etc.), it is possible to determine the most
appropriate sampling strategy (Guglielmi et al., 2022, 2023), to reach the specific scientific objectives of each
cruise.
The data presented here are available online on the Seanoe servor (https://doi.org/10.17882/102337) in a
file identifying version v1 and v2. The sources of the original datasets (doi) with the associated references are
listed in the Supplementary Material (Tables S3, S4). As for version v1 we invite the users to comment on any
anomaly that would have not been detected or to suggest potential misqualification of data in the present product
(e.g. data probably good although assigned with flag 3, probably wrong). As for SOCAT or GLODAP, we
expect to update the SNAPO-CO2 dataset once new observations are obtained and controlled.
**7 Data availability**
Data presented in this study are available at Seanoe (Metzl et al, 2024d, https://www.seanoe.org,
https://doi.org/10.17882/102337. See also https://doi.org/10.17882/95414 for version V1. The dataset is also
available at https://explore.webodv.awi.de/ocean/carbon/snapo-co2/
*Author contributions*. NM prepared the data synthesis, the figures and wrote the draft of the manuscript with
contributions from all authors. JF measured the discrete samples since 2014, with the help from CM and CLM,
and prepared the individual reports for each project. NM and JF pre-qualified the discrete $A_T/C_T$ data. CLM and
NM are co-Is of the ongoing OISO project and qualified the underway $A_T/C_T$ data from OISO cruises. FT and
CG were PIs of the MINERVE cruises. All authors have contributed either to organizing cruises, sample
collection and/or data qualification, and reviewed the manuscript.
*Competing interest.* The authors have the following competing interests: At least one of the (co-)authors is a
member of the editorial board of Earth System Science Data



*Acknowledgments.* Most of the $A_T$ and $C_T$ data presented in this study were measured at the SNAPO-CO2 facility (Service National d'Analyse des Paramètres Océaniques du CO2) housed by the LOCEAN laboratory and part of the OSU ECCE Terra at Sorbonne University and INSU/CNRS analytical services. Support by INSU/CNRS, by OSU ECCE Terra and by LOCEAN, is gratefully acknowledged as well as support by different French "Services nationaux d'Observations", such as OISO/CARAUS, SOMLIT, PIRATA, SSS and MOOSE. We thank the research infrastructure ICOS (Integrated Carbon Observation System) France for funding a large part of the analyses. We thank the IRD (Institut de Recherche pour le Développement) and the French-Brazilian IRD-FAPEMA program for funding observations in the tropical Atlantic. We thank the French oceanographic fleet ("Flotte océanographique française") for financial and logistic support for most cruises listed in this synthesis and for the OISO program (https://campagnes.flotteoceanographique.fr/series/228/). We acknowledge the MOOSE program (Mediterranean Ocean Observing System for the Environment, https://campagnes.flotteoceanographique.fr/series/235/fr/) coordinated by CNRS-INSU and the Research Infrastructure ILICO (CNRS-IFREMER). The CocoriCO2 project was founded by European Maritime and Fisheries Fund (grant no. 344, 2020–2023) and benefited from a subsidy from the Adour-Garonne water agency. We thank the following programs coordinated by A. Tribollet which have contributed to the acquisition of the data in Mayotte: CARBODISS funded by CNRS-INSU in 2018-2019, Future Maore reefs funded by Next Generation UE-France Relance in 2021-2023, and OA-ME funded by a Belmont Forum International (ANR) in 2020-2026. We thank the program Mermex-Mistrals CNRS for supporting AMOR-BFlux, CARBORHONE, DICASE and MESURHOBENT cruises and the program EC2CO-INSU for supporting MISSRHODIA2 cruise. The ACCESS project was supported by CNRS MISTRALS and the DELTARHONE-1 by EC2CO-INSU. The ACIDHYPO project was founded by CNRS International Emerging Actions; we thank Captain and crew of the R/V Savannah from the Skidaway Institute of Oceanography (University of Georgia) for their support and technical assistance during the operations at sea. The AMAZOMIX project was funded by French Oceanographic Fleet, INSU (LEFE), IRD (LMI TAPICOA), CNES (TOSCA MIAMAZ project) and by the French-Brazilian international program GUYAMAZON. The OISO program was supported by the French institutes INSU (Institut National des Sciences de l'Univers) and IPEV (Institut Polaire Paul-Emile Victor), OSU Ecce-Terra (at Sorbonne Université), and the French program SOERE/Great-Gases. We also thank the Research Infrastructure ILICO (https://www.ir-ilico.fr). We warmly thank Alain Poisson who initiated the MINERVE program and performed many of the measurements onboard R/V Astrolabe from 2002 through 2018. We thank all colleagues and students who participated to the cruises and have carefully collected the precious seawater samples. We thank Frédéric Merceur (IFREMER) for preparing the page and data availability on Seanoe and Reiner Schlitzer (AWI) for including the SNAPO-CO2 dataset in the ODV portal.

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
