# Peer review of "An updated synthesis of ocean total alkalinity and dissolved inorganic carbon measurements"

_Earth System Science Data, 2024_

## Author Comment (AC1)

Response to Reviewers' comments on the manuscript:
An updated synthesis of ocean total alkalinity and dissolved inorganic carbon measurements from 1993 to 2023: the SNAPO-CO2-v2 dataset, MS No.: essd-2024-464

Reply to Reviewer 1, Kim Currie (in black from reviewer, in blue our reply)

General Comments

Marine carbon data that are of high and known quality, have appropriate metadata and are FAIR are vital in assessing the role of the ocean in the changing carbon cycle. The SNAPO-CO2- v2 alkalinity and DIC dataset described in this manuscript is an important contribution as the data have wide spatio and temporal coverage and are of very high quality. The v2 described in this manuscript builds on the initial dataset published earlier this year. The manuscript details the data sources, the QC processing, and then the data assemblage and synthesis. Five use-cases are then presented, with the spatial distribution and trend analysis of the five cases assessed.

The data are available via the SEANOE website, and via the doi number presented in the manuscript.

The paper is very well written, the tables and diagrams are clear, except where noted below, and the supplementary material is useful and suitable.

This is a very good manuscript, I'm sure the dataset will be well used by the marine carbon community and that the manuscript will be well cited. I recommend publication and have only a few minor comments outlined below.

Response: We warmly thank Kim Currie for her support.

Specific Comments

Line 433        missing word: "…precision was based on triplicate analyses was estimated.…"

Response: Thank you: sentence revised as follow:
"The AT precision of ± 2 µmol kg-1 was based on triplicate analyses (Lagoutte et al., 2023)."

Figure 7, shows the surface distribution of alkalinity and DIC in the Mediterranean Sea over the decade 2014 – 2023. Given that the trend in DIC over that decade was 0.72 umol kg-1 yr-1 , the changes in the graphed value could be up to 7.2 umol kg-1 for the decade. I suggest either normalizing the data to a particular year, or to describe the effect in the text.

Response: Thank you for this comment. In this version, as an example of the results in the Mediterranean Sea we have presented a map of AT and CT in surface waters for the period 2014-2023. In the previous paper (SNAPO-CO2-v1) a map for the period 1998–2019 was presented using all data available with the aim at showing the distribution at regional scale. In this new version, we preferred to change the period, 2014-2023, because we added new data and we selected the data for one decade. As noted by the reviewer, the CT trend is positive, but this was obtained at one location around DYFAMED time-series. Unfortunately, we have no information of the trend in all sectors and it is not possible to extrapolate at basin scale as the assumption that the trend at one location is representative of the whole basin cannot be argumented for. For the period 2014-2023, the observed trend is 1.02 µmol/kg/yr against 0.72 µmol/kg/yr for the full period (1998-2023) as shown in Figure 8 or 0.94 µmol/kg/yr and 1.53 µmol/kg/yr for different seasons (as listed in Table 7). At subsurface, the 1.1 µmol/kg/yr trend probably mimics the CT increase due to anthropogenic CO2 (as discussed in the text). As suggested by the reviewer, we thus attempted to normalize the data

presented in figure 7. As a test, we normalized the data for year 2019 (when a MOOSE-GE cruise was conducted in June) and using a trend of +1 µmol/kg/yr applied to the CT data. The new map with normalized CT (noted CT-ref) is presented below (Figure R1) showing the same distribution compared to the map deduced from the original observations.

[Figure]

Figure R1: Distribution of CT (µmol kg-1) in surface waters of the Mediterranean Sea (0-10m) from observations in 2014-2023 (left, as in Figure 7b in the MS) and when CT is normalized to the reference year 2019 (CT-ref, on right). Because the corrections range from -6 to 4 µmol kg-1, the distribution is almost the same.

For clarity we have added the sentence:

"Note that, given the observed CT trends, the spatial view presented in Figure 7b for 2014-2023 would be the same based on CT concentrations normalized to a reference year (not shown)."

Several of the Case studies in Section 5 involve time series analysis, however the methodology for doing this is not described or referenced. Sutton et al (2022) is a useful reference for this:

Sutton, A.J., Battisti, R., Carter, B., Evans, W., Newton, J., Alin, S., Bates, N.R., Cai, W.-J., Currie, K., Feely, R.A., Sabine, C., Tanhua, T., Tilbrook, B., Wanninkhof, R. (2022) Advancing best practices for assessing trends of ocean acidification time series. Frontiers in Marine Science, 9: 1045667. doi: 10.3389/fmars.2022.1045667

Response: Thank you for this comment. In section 5 we have presented a few examples of trend analysis based on data for different periods and seasons. For all results (Table 7) we have calculated linear trends for selected data and mean values. Unfortunately the data are not regularly available (e.g. at monthly scale) and we cannot de-seasonalize the data as recommended (Sutton et al, 2022).

This is now specified and we added the suggested reference as follows when introducing section 5:

"5 Regional AT and CT distributions and trends based on the SNAPO-CO2 dataset

The regional distributions are described for the Mediterranean Sea and for selected regions in the open ocean and coastal zones where the data are available for 10 years or more to explore the AT and CT trends. Given the observed seasonal and inter-annual variability and that the time-series were not regular (e.g. at monthly frequency), we cannot use recommended methods to estimate the trends (e.g. based on de-seasoned data, Sutton et al, 2022). Here we have selected the locations and seasons where the CT trends can be linearly fitted and compared with no interpolation to fill gaps and discontinuous data (e.g., fewer samples during the COVID period)."

Line 930        This dataset is indeed a useful complement to other data compilations such as GLODAP and SOCAT.  Are these SNAPO-CO2 data included in SOCAT and / or GLODAP (as appropriate)?

Response: This is an important suggestion, however SOCAT includes only fCO2 and GLODAP includes other properties (e.g. O2, nutrients). When SOCAT was first discussed (in 2007), we wanted to include any carbonate system property (pCO2, AT, CT, pH), but it was decided to include only fCO2 data (see also our reply to reviewer 2). Note also that in a paper in preparation (Jiang et al, in prep) information on SNAPO-CO2 along with GLODAP, SOCAT and many other data-products are synthetized. Users can thus merge some of these products for specific studies.

Figure 14. The colour scale is not necessary, and is not used in the Figure data. There are only two time periods identified by the colours, and these are indicated in the caption.

Response: Thank you, figure 14 corrected.

Fig S3 The caption is incorrect, the colour coding in the Figure is not as described in the caption. This needs to be corrected.

Response: Thank you, figure S3 corrected.

Fig S4 The concept of this Figure is good, and it relates well to the text (line 339), however the map insert is difficult to read, and it is difficult to relate the colour coding of the triangle symbols in the vertical profiles with the day scale on the map. This should be clarified, - if the time component is important the colours or symbols should be the same, otherwise the day scale is not necessary.

Response: Thank you. The aim was to show results for 2 cruises conducted in the same region and same period. The day scale is not necessary. Figure S4 has been corrected (Figure R2) accordingly (new colors for symbols to highlight the difference at depth for AMAZOMIX and TARA cruises).

[Figure]

Figure R2 (for revised figure S4): Profiles of AT and CT in the western tropical Atlantic (near the Amazon River plume) for 2 cruises conducted in September 2021 (Stations from AMAZOMIX on 15-22-Sept-2021 and TARA-Microbiome on 1-Sept-2021). The location of the selected stations are shown in the inserted map (TARA-Microbiome in red, AMAZOMIX in blue) produced with ODV (Schlitzer, 2018). Mean values of the properties at 1000m are listed in Table 4 (main text).

Reference in this reply:

Jiang, L.Q., Fay, A., Müller, J. D. et al: Synthesis products for ocean carbon chemistry. In prep., 2024.

---

## Author Comment (AC3)

Response to Reviewers' comments on the manuscript:
An updated synthesis of ocean total alkalinity and dissolved inorganic carbon measurements from 1993 to 2023: the SNAPO-CO2-v2 dataset, MS No.: essd-2024-464

Reply to Reviewer 2, Toste Tanhua (in black from reviewer, in blue our reply)

The manuscript describes a data set of about 67000 observations of ocean dissolved inorganic carbon and/or total alkalinity around the world. The data set is mostly based on observations by the French vessels and scientists, analysed in a lab in France.

This is a valuable compilation of data in a single format and with a coherent quality control. The manuscript is well written and perceived. Quality controlled data in a consistent format of ocean carbon variables are valuable, so the manuscript deserves to be published.

The ms refers to both the GLODAP and SOCAT data products. There is some overlap, but also differences. In particular for GLODAP there is a potential overlap since that is also dealing with interior ocean DIC and TA data. It would be good if the ms could state how large a fraction of the SNAPO data are already in GLODAP, and an estimate on how large a fraction will be submitted to GLODAP for future versions.

Very often when using ocean carbon data, there is a need and s strong correlation to other variables. I can imagine that often (but probably far from always) other variables were being measured during these campaigns, fgor instance, oxygen and nutrients (variables often needed to calculate the anthropogenic component of the DIC). However, by looking at the individual data sets for those cruises where that is available, I was in most cases not able to locate the other variables, or even find a list of other variables that could be available. I realize that amassing other variables as well, just as is done in GLODAP or SPOTS, for instance, is probably outside the scope of this work. However, it would be useful to have that information about additional variable available in a concise format, for instance in tables S1. At least for the "most important auxiliary variables", possibly guided by variables available in GLODAP.

Response: We warmly thank Toste Tanhua for his support.

The reviewer is correct, some (but only a small proportion) data of the SNAPO-CO2 synthesis are in GLODAP (e.g., OUTPACE, PANDORA, EGEE, BIOZAIRE) or SPOTS (e.g., DYFAMED). The SNAPO-CO2 dataset is dedicated to AT and CT data, somehow like SOCAT for fCO2.  When we started SOCAT (at a workshop in Paris UNESCO, 2007) it was suggested to include not only fCO2 data but also AT, CT or pH. However, due to available personnel and technical issues, only fCO2 was selected for quality control in SOCAT (which is a very important step anyway). Here, we decided to start a synthesis of AT-CT dataset with the data we have on hand and measured with the same technic. This may motivate other groups to do the same and maybe a way to start SODAT… (Surface Ocean Dic AT data) ?

For other properties, if available (e.g. for anthropogenic estimates or associate nutrients), users interested can find information in the DOI listed in Table S3. In addition, on the Seanoe page where the SNAPO-CO2 data is archived (https://doi.org/10.17882/102337), there is a list of the projects and their link. As an example, a user interested with the North Atlantic can obtain the data from SURATLANT at Seanoe (see Reverdin et al, 2018, 2023, https://doi.org/10.17882/54517). As noted in the manuscript, we encourage users to contact the PIs (listed in Table S1a and S1b) to get information on all properties measured for each project.

Note also that in a paper in preparation (Jiang et al, in prep) information on SNAPO-CO2 along with GLODAP, SOCAT and many other data-products are synthetized. Users can thus merge some of these products for specific study. Finally, as was done for SNAPO-CO2-v1, the SNAPO-CO2-v2 dataset will be available in GOA-ON for SDG 14.3.1 (https://oa.iode.org/); this is specified line 82 in the submitted MS. The data will be also available in ODV (see line 1009: https://explore.webodv.awi.de/ocean/carbon/snapo-co2/).

Reference in this reply:

Jiang, L.Q., Fay, A., Müller, J. D. et al: Synthesis products for ocean carbon chemistry. In prep., 2024.

Reverdin, G., Metzl, N., Olafsdottir, S., Racapé, V., Takahashi, T., Benetti, M., Valdimarsson, H., Benoit-Cattin, A., Danielsen, M., Fin, J., Naamar, A., Pierrot, D., Sullivan, K., Bringas, F., and Goni, G.: SURATLANT: a 1993–2017 surface sampling in the central part of the North Atlantic subpolar gyre, Earth Syst. Sci. Data, 10, 1901-1924, https://doi.org/10.5194/essd-10-1901-2018, 2018.

Reverdin, G., Metzl, N., Olafsdottir, S., Racapé, V., Takahashi, T., Benetti, M., Valdimarsson, H., Quay, P. D., Benoit-Cattin, A., Danielsen, M., Fin, J., Naamar, A., Pierrot, D., Sullivan, K., Bringas, F., Goni, G., Becker M., Leseurre C., and Olsen A.: SURATLANT: a surface dataset in the central part of the North Atlantic subpolar gyre. SEANOE. https://doi.org/10.17882/54517, 2023.

---

## Author Response (AR2)

Reply to comment from Editor

;;;;;;; Mail from Editor:

Dear dr. Metzl,

Given the positive reviews of both referees, I am happy to accept your article for publication.
I have only one minor request - throughout the manuscript (at L604, L681, L748, L848) you mention 'not shown'.
I would ask you to omit that at L604, as it is redundant, but at the other instances where you discuss time trends, I would ask you to either provide a reference for the statements, or provide a figure. The reader should be able to evaluate how important/significant the trends are that you mention.

Kind regards
Sebastiaan

;;;;;;;;;

Reply to the editor:

Dear Editor, dear Sebastiaan

Thank you for your positive report and suggestions for the « not shown » sections.
The text has been revised accordingly and appropriate figures (3 figures) added in the supplementary material.

Best regards,

Nicolas Metzl

;;;;;;;

---

## Author Response (AR3)

Note to the editor, 22 January 2025.

In the final version, I have changed the table numbers (but not the tables). Text, tables, figures and SuppMat are the same as the version accepted.

Regards,

Nicolas Metzl